# Integrated information structure collapses with anesthetic loss of conscious arousal in *Drosophila melanogaster*

**Angus Leung** [1] *, **Dror Cohen** [1,2], **Bruno van Swinderen** [3], **Naotsugu Tsuchiya** [1,2,4,5] *

**1** School of Psychological Sciences, Monash University, Melbourne, Australia, **2** Center for Information and Neural Networks (CiNet), National Institute of Information and Communications Technology (NICT), Osaka, Japan, **3** Queensland Brain Institute, The University of Queensland, Brisbane, Australia, **4** Monash Institute of Cognitive and Clinical Neuroscience (MICCN), Monash University, Melbourne, Australia, **5** Advanced Telecommunications Research Computational Neuroscience Laboratories, Kyoto, Japan

* angus.leung1@monash.edu (AL); naotsugu.tsuchiya@monash.edu (NT)

**Data Availability Statement:** Preprocessed fly LFPs from which information structures were constructed are available on Figshare - doi: 10. 26180/5ebe420ae8d89.

## Abstract

The physical basis of consciousness remains one of the most elusive concepts in current science. One influential conjecture is that consciousness is to do with some form of causality, measurable through information. The integrated information theory of consciousness (IIT) proposes that conscious experience, filled with rich and specific content, corresponds directly to a hierarchically organised, irreducible pattern of causal interactions; i.e. an integrated informational structure among elements of a system. Here, we tested this conjecture in a simple biological system (fruit flies), estimating the information structure of the system during wakefulness and general anesthesia. Consistent with this conjecture, we found that integrated interactions among populations of neurons during wakefulness collapsed to isolated clusters of interactions during anesthesia. We used classification analysis to quantify the accuracy of discrimination between wakeful and anesthetised states, and found that informational structures inferred conscious states with greater accuracy than a scalar summary of the structure, a measure which is generally championed as the main measure of IIT. In stark contrast to a view which assumes feedforward architecture for insect brains, especially fly visual systems, we found rich information structures, which cannot arise from purely feedforward systems, occurred across the fly brain. Further, these information structures collapsed uniformly across the brain during anesthesia. Our results speak to the potential utility of the novel concept of an "informational structure" as a measure for level of consciousness, above and beyond simple scalar values.

## Author summary

The physical basis of consciousness remains elusive. Efforts to measure consciousness have generally been restricted to simple, scalar quantities which summarise the complexity of a system, inspired by integrated information theory, which links a multi-dimensional, informational structure to the contents of experience in a system. Due to the complexity

**Funding:** AL was supported by an Australian Government Research Training Program (RTP) Scholarship. DC was supported by a JSPS international postdoctoral research fellowship and JSPS KAKENHI 18F18706. DC and NT were supported by TWCF0199 from Templeton World Charity Foundation, Inc. NT was funded by the Australian Research Council Future Fellowship FT120100619 and the Discovery Project DP180104128 and DP180100396 and the National Health and Medical Research Council APP1183280. BvS was funded by the National Health and Medical Research Council Project APP1103923. The funders had no role in study design, data collection and analysis, decision to publish, or preparation of the manuscript.

**Competing interests:** The authors have declared that no competing interests exist.

of the definition of the structure, assessment of its utility as a measure of conscious arousal in a system has largely been ignored. In this manuscript we evaluate the utility of such an information structure in measuring the level of arousal in the fruit fly. Our results indicate that this structure can be more informative about the level of arousal in a system than even the single-value summary proposed by the theory itself. These results may push consciousness research towards the notion of multi-dimensional informational structures, instead of traditional scalar summaries.

## Introduction

The question of how subjective, conscious experience arises from physical interactions has been pondered by philosophers for centuries [1,2], and now has moved into the domain of cognitive neuroscience [3–5]. Because we are only able to experience our own individual consciousness, exact inference of others' conscious contents (i.e., what it is like to be a bat [1]) seems intractable. However, broader inference on levels of consciousness, ranging from low during coma and deep anesthesia to high in wakeful states seems possible across animals. Behaviors of animals, ranging from humans to insects, all seem to change in a similar manner from highly active wakefulness with marked high-level cognitive capability to loss of consciousness with negligible cognitive functions. Indeed, such inferences have been widely accepted across various losses of consciousness in brain damaged patients [6] and non-human mammals [7,8], and are now becoming applied to insects [9–12].

As stated, complex behavioural repertoires of animals, ranging from humans to insects, all seem to reduce in a similar manner from highly active wakefulness to loss of consciousness. During wakefulness, flies, for instance, have been shown to exhibit processes such as working memory [13–15], attention [16–18], and feature binding [19]. Flies also seem to experience varying states of arousal which are physiologically regulated in a similar manner to mammals, such as sleep [9,20,21] and anesthesia [11,12]. Despite these similarities, processing in the fly brain is largely thought to be feedforward, with potential exception of central structures such as the central complex and mushroom bodies [22–24]. This is in contrast to a more integrative view of the seemingly more complex brains of mammals, featuring both feedforward and feedback interactions during wakefulness from primary sensory areas to midbrain and executive areas [25–27].

The importance of feedback for conscious processing is emphasised in an influential view that consciousness arises with "integrated information" [28–30]. Integrated information, distinct from the standard notion of Shannon information [31], is defined as "differences that make a difference within a system" [32,29,30]. In other words, integrated information is concerned with how elements of a system causally influence each other such that information is accessible to the system itself (extrinsic information, conversely, concerns how states of a system causally influence states of another, separate system; see supporting information in [33]). Integrated information theory (IIT; [28,30,34]) provides a mathematical quantification of integrated information, and proposes that it is critical for consciousness to arise. Specifically, IIT describes how hierarchically organized elements uniquely and causally interact with other elements within a system in an integrated manner to produce information accessible to the system itself. According to IIT, the "maximally irreducible conceptual structure" [30] is hypothesised to directly correspond to the quantity and quality of consciousness. That is, the richer and more specific the informational structure of the system, the higher the level of consciousness in a system, and the richer the contents that the system consciously experiences.

Critically, the hierarchically organised elements must both exert effects on other elements and receive effects from others, all within the system, and thus these structures can only arise with the presence of both feedforward and feedback interactions.

While IIT offers a compelling theoretical account linking integrated information and consciousness, empirical applications of the theory remain rare [35]. Thus, whether *empirically* estimated integrated information structures relate to conscious arousal remains largely unknown. While we as yet cannot be certain of consciousness in flies, they pose an interesting system to apply the theory. In particular, regardless of consciousness per se, a purely feedforward brain should give zero integrated information and correspondingly a minimal informational structure. Thus, we address the following questions. First, how can we estimate informational structures from neural activity recorded from a biological system? Second, does the fly brain generate integrated information and non-minimal information structures? If it does, would the structures be collapsed during reduced arousal as manipulated using general anesthesia? And third, does integrated information and its associated information structures arise (and subsequently collapse during anesthesia) primarily in the central regions of the fly brain?

We address the above questions by analyzing neural recordings from the fruit fly, collected during wakefulness and isoflurane anesthesia [12,36]. We apply a novel construct, "integrated information structures" (IIS), to capture the level of arousal of the fly. We found that the structures which were present during wakefulness collapsed during anesthesia. Critically, they were better at classifying arousal states than a scalar summary (i.e. just a single number), a measure which is usually championed as "integrated information" in IIT, with their collapse occurring all throughout the fly brain. Our results indicate the presence of feedback interactions across the fly brain during wakefulness, and demonstrate the utility of information structure as a measure for level of arousal, above and beyond simple scalar values, opening the door for improved clinical measures of consciousness.

## Results

### Constructing integrated information structures from fly local field potentials

To construct the IIS, we used local field potentials (LFPs; hereafter referred to as "channels") recorded from the fruit fly brain (Fig 1A; see Methods; [12]). LFPs were recorded using a linear multi-electrode array, such that 15 channels covered both peripheral and central regions of the brain. We operationally defined the discrete state of each channel at each time by binarizing it with respect to the median voltage of that channel (Fig 1B; see also S1 Text for effect of binarizing using different thresholds).

Fig 1C–1I illustrates the steps to estimate the IIS of two channels, A and B. From the empirically observed time course of the two discretized channels, we first construct a transition probability matrix (TPM; Fig 1C). Each entry of the TPM gives the probability of a given channel taking some state in the future, given the current state of all channels in the system (see Methods). Thus, the TPM characterizes how the whole system (A and B) evolves over time, containing all necessary information for unfolding how subsets of the system (A, B, and AB) "causally" (in a statistical sense) interact to irreducibly specify the state of the whole system (AB). We refer to causality as statistically inferred from conditional probability distributions [37], which is not necessarily the same as perturbational causality [38]. We return to the issue of estimating the TPM from observed versus perturbed transitions in the Discussion. Importantly, we use the TPM to measure the information that each subset of the system specifies regarding some other subset of the system, as we describe below.

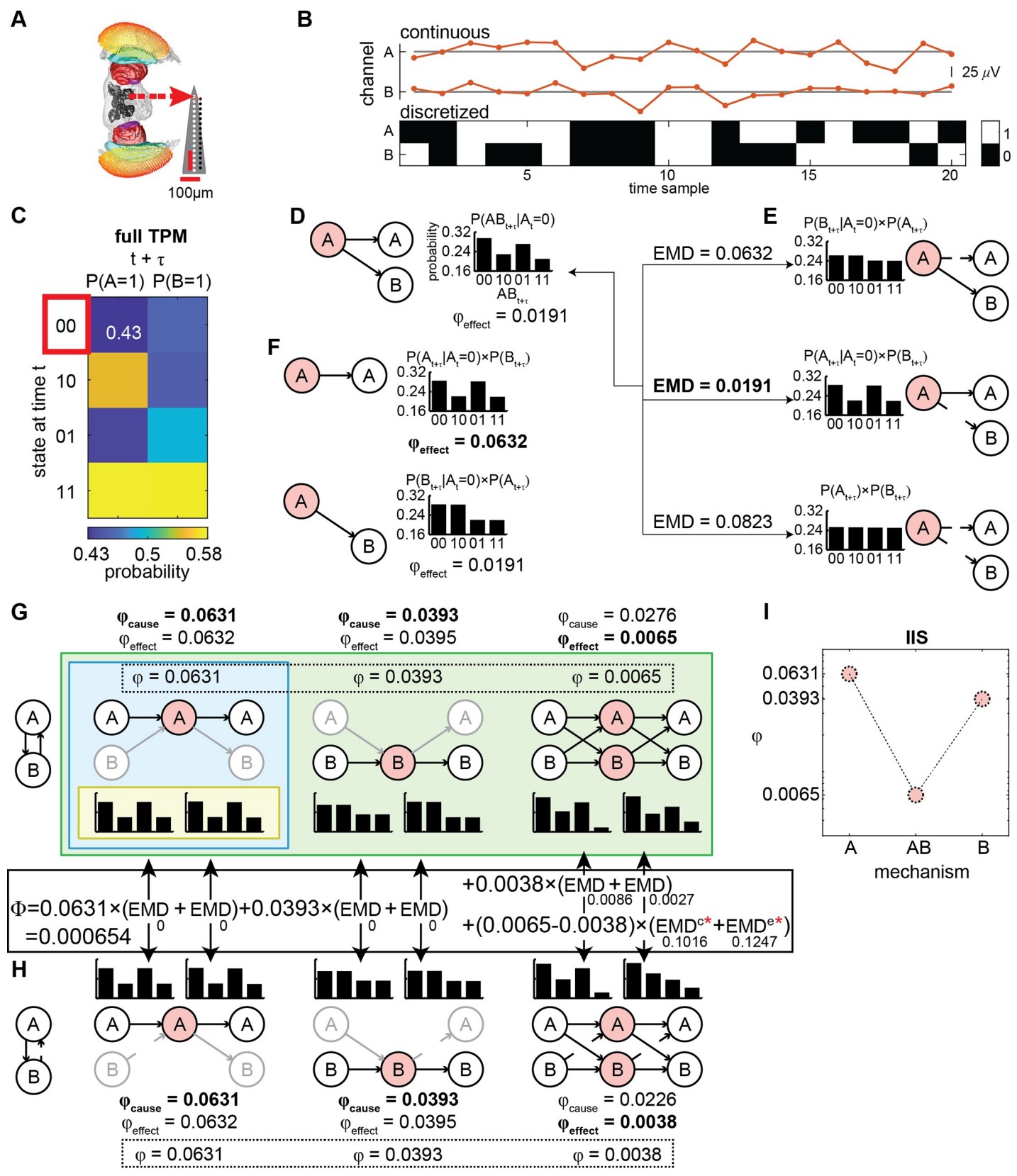

**Fig 1. Summary of IIT and processing pipeline for computing the IIS from LFPs.** (**A**) Multi-electrode probe recording of LFPs from the fly. (**B**) Continuous LFPs (red, top) are discretized (black/white, bottom) by comparing to the median voltage for each trial. Displayed is an example of 20 samples for a set of two channels A and B. (**C**) A state-by-channel transition probability matrix (TPM; see Methods) describes how the state of a system at time $t$ specifies the possible future states of each channel at time $t+\tau$ ($\tau = 4$ ms). For example, the top left entry of the full TPM is 0.43, which represents the probability of channel A being '1' at time $t+\tau$ given that channel A and B were both '0' at time $t$. (**D**) At a given state (e.g. A = '0' and B = '0' at time $t$, outlined in red in **C**, the effect information specified by a subset ("mechanism"; here A, in light red) over the future states of another subset ("purview"; here A and B, in white), is given by the probability distribution of the purview conditioned on the current state of the mechanism. (**E**) To compute integrated information ($\varphi_{effect}$) of mechanism A over purview AB, we find the disconnections (i.e. replacing connections with random-noise connections) between the mechanism and the purview (indicated by broken arrows) which best approximate the original probability distribution. We compare the disconnected probability distributions to the original distribution using the earth mover's distance (EMD; treating probabilities as "earth" to be moved). We interpret the minimum EMD (bolded) as irreducible information generated over the purview by the mechanism (i.e. $\varphi_{effect}$). (**F**) We compute $\varphi_{effect}$ for every possible purview (A, B, and AB as in **D**, with values 0.0632, 0.0191, and 0.0191 respectively), and select the purview and its associated probability distribution which gives the maximally integrated effect (bold). As probability distributions (bar graphs), we display the distribution over both channels A and B, assuming the maximum entropy distribution and independence on the channels outside of the purview. (**G**) $\varphi_{cause}$ is determined in the same manner as $\varphi_{effect}$, except looking at possible past states of the purview (at $t-\tau$). Both $\varphi_{cause}$ and $\varphi_{effect}$, and their associated probability distributions, are determined for every mechanism (A, B, and AB; left and right distributions are cause and effect probability distributions of the selected purviews; channels outside of the purview are greyed out). The overall $\varphi$ generated by a mechanism is the minimum of $\varphi_{cause}$ and $\varphi_{effect}$ (bolded and in the dotted box). Yellow, blue, and green backgrounds (innermost, middle, and outermost rectangles) indicate correspondence with the IIT terminology of "cause-effect repertoire", "concept", and "cause-effect structure" (CES), respectively. (**H**) All $\varphi$ values and associated probability distributions are re-computed for each possible uni-directional cut (again, replacing with random-noise connections) separating the channels into a feedforward interaction from one subset of channels to the remainder of the system. Broken lines here depict the cut removing channel B's input to A. System-level integrated information ($\Phi$) is the sum of distances between cause and effect probability distributions specified by the full and (minimally) disconnected system, weighted by the $\varphi$ value for each mechanism (hence $\Phi$ is the minimum across all possible system level cuts; solid box between **G** and **H**; see Methods for details on $EMD^c$ and $EMD^e$ which are marked by red asterisks). Note that distances between 1-channel mechanisms were 0, not contributing to $\Phi$, which we found to be the case in general (see also S6 Text about the role of 1-channel mechanism in our results). (**I**) We take the $\varphi$ values of each mechanism (within the dotted box in **G**) to form the integrated information structure (IIS).

Fig 1D considers how subset A's current state (A = '0' at time $t$) specifies the future state of any subset of the system AB (at time $t+\tau$; we use $\tau = 4$ ms; we repeated analyses also at $\tau = 2$ ms and 6 ms, see S2 Text). For brevity we will refer to the subset whose current status is analyzed (red circles in Fig 1D–1G) as a "mechanism", and the affected subset (white circles in Fig 1D–1G) as a "purview", following IIT terminology; [30]. Based on the TPM, we can compute a probability distribution over past and future purview states (the bar graphs in Fig 1D–1H), given the current state of the mechanism in consideration; for example, the bar graph in Fig 1D shows that if mechanism A is in state '0' at time $t$, AB is more likely to be '00' or '01' than '10' or '11' at time $t+\tau$. Such a probability distribution specifies the information generated by a mechanism over a given purview.

Fig 1E illustrates the procedure to find "irreducibility" of the causal interaction from mechanism A to purview AB. To estimate how much the purview is irreducible, or uniquely determined by integrative interactions between A and the purview (according to IIT's integration axiom), we estimate probability distributions assuming that some causal interactions are "disconnected" (i.e. statistically noised; see S3 Text). We quantify the degree of causal interactions by computing the distance between the two probability distributions (distance is measured using the earth mover's distance, with probabilities being moved as "earth"; EMD; [39]). The distance between the full (Fig 1D) and disconnected distribution which best approximates (i.e. is closest to) the original full distribution (Fig 1E) quantifies integrated information $\varphi$. Here, $\varphi$ of A on AB can be understood as the degree to which mechanism A generates information about purview AB, above and beyond independent parts. In Fig 1E, the disconnection from A to B minimally affects the distribution out of all the possible cuts, giving $\varphi$ of 0.0191.

Next, Fig 1F illustrates the identification of the purview over which A generates the most integrated information, as dictated by IIT's exclusion axiom (the exclusion axiom in this context means that only the maximal information specified by A should be considered in order to avoid information being multiplied beyond necessity). The purview for which A generates the most integrated information is referred to as A's "core effect". Here, mechanism A has a set of candidate purviews: A, B and AB. Based on the current state of A ('0'), we repeat the process of measuring distances between full distributions and disconnected distributions among all

purviews (i.e. all subsets which are potentially affected by A). In this particular case, purview A is the core effect (φ of A on A is 0.0632, compared to φ of A on B and φ of A on AB both being 0.0191). Next, we perform similar operations on the TPM, but now looking at information the mechanism generates about a purview's *past*, instead of future. This is done to estimate the core *cause* of A. According to the intrinsic existence axiom of IIT, we consider A's overall influence (i.e. the information it generates for the system), to be the minimum of A's cause and effect. Consequently, a mechanism which only provides outputs to its purview (i.e. only specifies its effects), or only takes inputs from its purview (i.e. only specifies causes), generates zero integrated information.

Repeating the procedure (Fig 1D–1F) for all candidate mechanisms (A, B, and AB), Fig 1G characterizes how all possible elements of the system specify the set of structured and integrated causal interactions, listing a full set of core causes and effects of all the mechanisms. The full set of distributions for all core causes and effects for all mechanisms, and their associated integrated information values is referred to in IIT as a cause-effect structure (CES).

Finally, Fig 1H explains how IIT arrives at a purported measure of level of consciousness, system-level integrated information Φ, through a system-level disconnection. The process of identifying core causes and effects for each mechanism is repeated after making unidirectional disconnections to the full system, in the same manner as disconnecting mechanisms from purviews. System-level integrated information is the sum of EMDs between the full CES and the CES of the statistically disconnected system, weighted by the integrated information φ of each mechanism in the full CES (as depicted in the calculation between Fig 1H and 1G; see Methods). Once again, as there are many possible ways of disconnecting the system, we select the disconnection which best approximates the CES of the full, whole system (i.e. which generates the smallest weighted EMD between the full CES and the disconnected CES). Consequently, a completely feedforward system generates zero system-level integrated information, as the unidirectional disconnection of feedback connections (which are actually non-existent) will yield identical probability distributions for all mechanisms and thus an identical CES as the fully connected system. In the case of the 2-channel system AB, the minimal disconnection is the disconnection from B to A. This disconnected CES is used to assess system-level integrated information (for details, see Methods).

One difficulty with Φ is the high computational cost due to the combinatorial explosion of all possible system cuts. To enable us to search through all possible cuts, we restricted analysis to 4 channels at a time, using every combination of 4 channels as a "system". This provided a good balance between spatial coverage for each set of channels and computation time.

We also considered a computationally cheaper alternative to Φ. Specifically, we assessed a set of φ values, which we term Integrated Information Structure (IIS; Fig 1I), as an alternative measure for discriminating level of consciousness. A set of mechanism-level φ values are faster to compute, as they are already obtained as part of the computation of Φ. The IIS is an approximation of the full cause-effect structure proposed by IIT [39]. While the cause-effect structure requires causal intervention for building the TPM, here we only observe interactions as they naturally occur over time. Further, the full cause-effect structure holds details beyond just integrated information values, specifically the purviews of each mechanism and their associated probability distributions, whereas for simplicity the IIS only considers the integrated information values themselves. As system-level integrated information and the IIS are obtained for each possible state of the system, we averaged across these states, weighting by the occurrences of each state [40].

Fig 2 shows an example IIS obtained from 1 fly, 1 channel set during both wakefulness and anesthesia, when extending this process to the 4-channel case. IIT provides two main hypotheses for this paper: 1) system-level integrated information (Φ) should be reduced by general

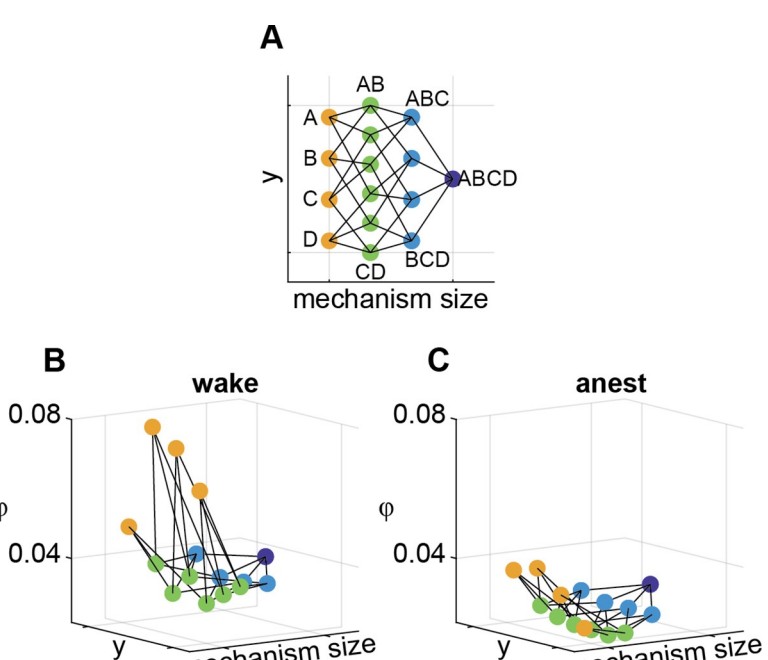

**Fig 2. A 3D representation of an integrated information structure (IIS) for one channel set for one fly.** (**A**) Top-down view of the IIS. Mechanism size refers to the number of channels that constitute each mechanism (yellow, green, light blue, and dark blue dots indicate mechanisms consisting of 1, 2, 3, and 4 channels respectively). The y-axis is arbitrarily set to give equal spacing between mechanisms. Lines indicate inclusion relations (e.g., mechanism AB consists of A and B). (**B**) An exemplar IIS from a single fly and channel set, during wakefulness. (**C**) An IIS from the same fly and channel set as in **B** during anesthesia. A 3D rotation video of the IIS is available at http://dx.doi.org/10.26180/5eb952457b48f.

anesthesia, and 2) a set of mechanism-level integrated information (φ) values, the IIS, should also collapse during general anesthesia, reflected by reduced φ values for each mechanism (as opposed to increased φ for some mechanisms). While IIT does not explicitly predict the latter, we reasoned that level of consciousness should generally correlate with the richness of contents of consciousness. Note that these hypotheses here cannot directly confirm or invalidate IIT as a theory of consciousness, as the nature of insect consciousness is still unclear, and we do not apply every aspect of IIT (due to feasibility issues), which we expand on in the Discussion.

## System-level integrated information reduces globally due to general anesthesia

We first checked the prediction that system-level integrated information (Φ), IIT's proposed measure of level of consciousness, was reduced during anesthesia. Using linear mixed effects analysis (to account for intra-fly channel set correlations; see Methods), we indeed found system-level integrated information to be significantly affected by anesthesia ($\chi^2(1) = 6.656 \times 10^3$, $p < .001$; likelihood ratio test, see Methods). Specifically, it was reduced during anesthesia (Figs 3 and 4A and 4B; β = -0.012, $t(12)$ = -2.525, $p$ = .013, one-tailed). This analysis also indicated that the fly LFPs did indeed generate non-minimal system-level integrated information during wakefulness. Across channel sets, system-level integrated information was significantly reduced during anesthesia for 12 of the 13 flies (S4 Text).

We next looked at whether anesthesia's effect depended on the spatial location of the channel sets. From the linear arrangement of channels from our recording setup, we characterised two features of each channel set, 1) the average location of channels in the set (relative to

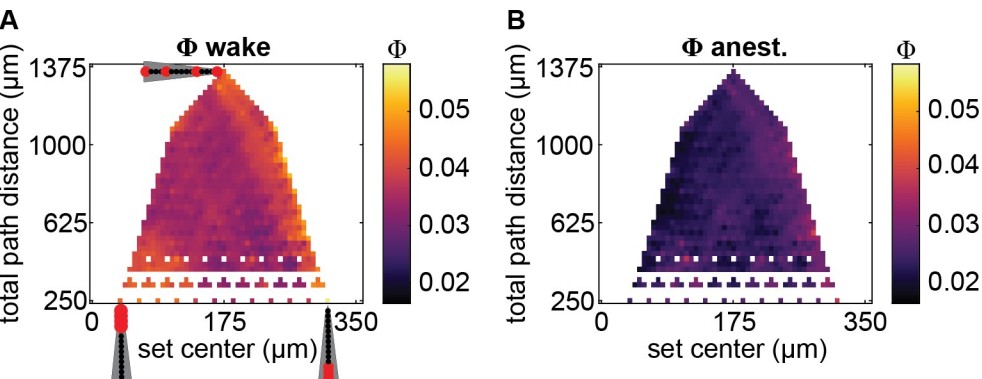

**Fig 3. Spatial map of system-level integrated information Φ.** (**A**) System-level integrated information Φ values during wakefulness, averaged across flies, as a function of average channel location relative to the position of the most central channel (x-axis; larger values indicate channel sets which on average are more peripherally located), and sum of pairwise distances between each pair of channels (total path distance; y-axis) within each channel set. Channel arrays, as in Fig 1A, indicate example locations of channels (in red) and their spacing along the two axes. Channel sets with identical centers and path distances were averaged. A subset of otherwise unfilled values in the map were linearly interpolated to reduce gaps in the map. (**B**) System-level integrated information Φ values during anesthesia.

channel 1, the most central channel) and 2) the distance among channels in the set (the sum of pairwise distances between channel labels; total path distance).

Given the general view of the insect brain being largely feedforward [22,23], with potential exception of central brain structures which are responsible for integrating inputs from the periphery [10,24], we expected to find greater system-level integrated information for more centrally located channel sets. Opposite to our expectation, however, we found a trend indicating that peripheral channel sets tended to have slightly but significantly greater system-level integrated information ($\beta = 1.750 \times 10^{-2}$, $\chi^2(1) = 39.31$, $p < .001$). This trend was stronger during anesthesia, as indicated by a significant interaction between channel set location and wake/anesthesia condition ($\beta = 2.613 \times 10^{-2}$, $\chi^2(1) = 43.80$, $p < .001$). Thus, centrally located channel sets seemed to be more affected by anesthesia, despite having less system-level integrated information than peripheral channels. The latter finding is consistent with a view that central brains are more critical and sensitive to the level of arousal.

We also considered the effect of the spacing of channels within each channel set. If local recurrent connections drive the generation of integrated information, more "local" channel sets consisting of closely located channels would have greater system-level integrated information. Conversely, if long range recurrent connections are more important, more "global" channel sets consisting of widely spaced channels would have greater system-level integrated information, reflecting integration across the whole brain. We found system-level integrated information to increase slightly with greater distance among channels ($\beta = 1.364 \times 10^{-3}$, $\chi^2(1) = 5.351$, $p < .021$), with the direction of the trend being reversed during anesthesia (significant interaction between anesthesia and channel distance; $\beta = -2.714 \times 10^{-3}$, $\chi^2(1) = 10.59$, $p < .001$). Thus, during wakefulness, the more global sampling of channels tended to yield larger system-level integrated information, while anesthesia disrupted this effect to some extent.

Overall, system-level integrated information was reduced regardless of spatial location or distance among channels, suggesting the presence of both feedforward and feedback interactions all across the fly brain. So, for analysis on the multi-dimensional IIS, we analysed all channel sets together without dividing into groups based on location or distance among channels.

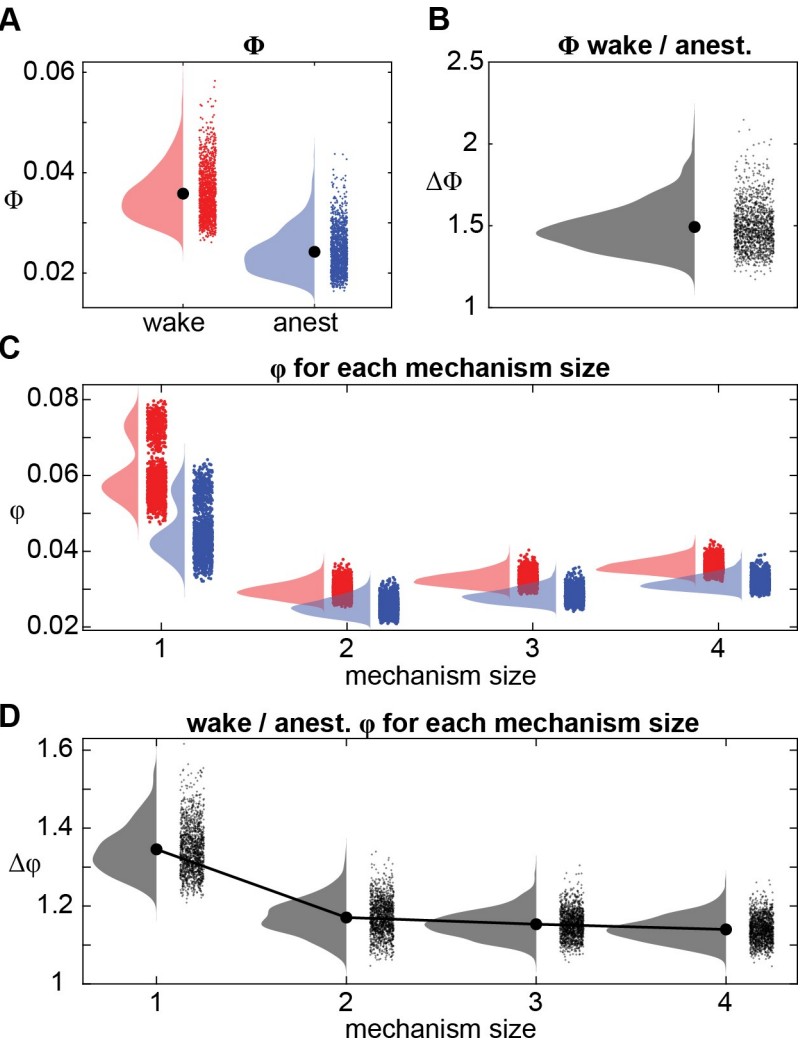

**Fig 4. Effect of anesthesia on system-level integrated information (Φ) and the integrated information structure (IIS: a set of φ values).** (**A**) Φ values during wakefulness (red) and anesthesia (blue) for each of 1365 channel sets, averaged across flies. (**B**) Ratio of Φ (wakeful / anesthetized), for all channel sets, averaged across flies. (**C**) φ values from the IIS for each mechanism size, for wake (red) and anesthesia (blue). We show the average value for each of 1365 channel sets averaged across flies for each mechanism size. (**D**) Ratio of wakeful φ to anesthetized φ (averaged across flies) for each mechanism size.

## Integrated information structure collapses due to general anesthesia

We next investigated integrated information (φ) for each mechanism during wakefulness, and compared them to those during anesthesia. First, we looked at the relationship between mechanism size and integrated information. Since larger mechanisms sample more sources of information, they have a greater capacity for integration, as compared to smaller mechanisms that sample fewer sources of information. Based on this, we reasoned that larger mechanisms will have greater integrated information.

We found integrated information values to significantly vary depending on the size of the mechanism ($\chi^2(3) = 1.512 \times 10^5$, $p < .001$; Fig 4C). Generally, we found that larger mechanisms generated greater integrated information (LME with two levels of mechanism size at a time, see Methods for details: 2-channel $<<$ 3-channel: $\beta = -2.941 \times 10^{-3}$, $t(12) = -18.73$, p $<$ .001; 3-channel $<<$ 4-channel: $\beta = -3.544 \times 10^{-3}$, $t(12) = -28.27$, p $<$ .001). However, 1-channel

mechanisms by far had the greatest integrated information overall (compared to 4-channel mechanisms: β = 0.025, $t(12)$ = 6.49, $p$ < .001). A potential explanation for the large difference in integrated information between 1-channel mechanisms and the other mechanisms is that 1-channel mechanisms are inherently irreducible to smaller parts. We return to this in the Discussion, offering other possible explanations.

Next, looking at the effect of anesthesia, we found integrated information to reduce significantly across all mechanisms with loss of arousal ($\chi^2(1)$ = 3.092 × 10$^4$, $p$ < .001; Fig 4C). We further found a significant interaction between anesthesia and mechanism size ($\chi^2(3)$ = 1.203 × 10$^4$, $p$ < .001), indicating that the extent to which integrated information was reduced due to anesthesia depended on mechanism sizes. We break down this interaction further in the next section.

## General anesthesia affects smaller mechanisms more than larger mechanisms

To understand the nature of the significant interaction between anesthesia and mechanism size, we next investigated how the different mechanism sizes were differentially affected by anesthesia. We expected that integrated information for larger mechanisms (consisting of more channels) would be affected more by anesthesia than smaller mechanisms. This is because anesthesia is known to preferentially disrupt global communication [12,36], and so its effect should be reflected more strongly in larger mechanisms involving many channels. To further illustrate, consider two pairs of strongly connected neurons, [AB and [CD], where there is a very weak connection between the two pairs (i.e., [AB]- -[CD]). In such a case, integrated information for both the 2-channel pairs ([AB] and [CD]) and the 4-channel mechanism ([ABCD]) could be high. If during anesthesia the connections between the pairs are disrupted, then 2-channel integrated information of the individual pairs could remain high while the overall 4-channel integrated information would reduce to zero.

To test if larger mechanisms were more greatly affected by anesthesia, we first analyzed the degree of reduction in integrated information as a function of mechanism size. To account for the variation in integrated information among mechanism sizes, we compared the ratio of wakeful to anesthetized integrated information. A larger ratio corresponds to a larger decrease in integrated information due to anesthesia. We verified that the ratio of wakeful to anesthetized integrated information was also significantly different among mechanism sizes ($\chi^2(3)$ = 2.229 × 10$^4$, p < .001; Fig 4D). However, instead of finding larger mechanisms to have larger relative reductions in integrated information due to anesthesia, we found the opposite—larger mechanisms had smaller relative reductions (β = 0.229, t(12) = 3.816, p = .003, β = 0.028, t(12) = 2.248, p = .044, and β = 0.017, t(12) = 2.444, p = .031, for comparing 1-, to 2-, 2- to 3-, and 3- to 4-channel mechanisms respectively). Given that the IIS indeed collapsed during anesthesia, we next sought to determine whether larger mechanisms better discriminated conscious level than smaller mechanisms, possibly because of lower variability or noise.

## Integrated information structure better distinguishes arousal level than system-level integrated information

Given that integrated information is reduced during anesthesia, we asked if this decrease is more reliable for larger mechanisms. We also sought to determine whether considering the entire IIS allows for better discrimination conscious level than just consideration of single mechanisms, i.e. is the pattern of integrated information useful above and beyond considering independent integrated information values in isolation? As IIT proposes the scalar system-level integrated information value as the measure of conscious level (whereas the multi-

dimensional IIS should represent experiential contents), we further compared this to the reliability of the decrease in system-level integrated information. While IIT touts system-level integrated information as a measure of conscious level, we reasoned that, as level of consciousness should generally correlate with the richness of its contents, the IIS would either match or even exceed the classification accuracy of system-level integrated information.

To compare the reliability of decreased integrated information, the collapse of the IIS, and decreased system-level integrated information, we used classification analysis. This allowed us to compare the reliability of one-dimensional changes of integrated information and system-level integrated information with multidimensional changes of the IIS. We used support vector machines (SVMs) to classify the conscious arousal level of individual epochs within each fly (within-fly classification, repeated for each fly; leave-one-paired-epoch-out cross-validation for each channel set; see Methods). To compare integrated information of different sized mechanisms, we averaged accuracies obtained across mechanisms of the same size.

We were able to discriminate wakefulness from anesthesia in the majority of channel sets, using either integrated information values or system-level integrated information (Fig 5A). Further, classification accuracy varied significantly depending on what measure was used (LME testing for main effect of mechanism size (1- to 4-channels), IIS, and system-level integrated information; $\chi^2(5) = 1.300 \times 10^4$, $p < .001$). Overall, 1-channel mechanisms achieved the greatest classification performance, significantly greater than 2-channel mechanisms ($\beta = 0.060$, $t(12) = 5.473$, $p < .001$) and 3-channel mechanisms ($\beta = 0.035$, $t(12) = 3.945$, $p = .002$), but not 4-channel mechanisms ($\beta = 0.018$, $t(12) = 2.033$, $p = .065$). Unexpectedly, integrated information of 1-channel mechanisms also matched that achieved by system-level integrated information, exceeding it slightly but not significantly so ($\beta = 0.008$, $t(12) = 0.5843$, $p = .570$). 1-channel mechanisms outperforming other mechanisms is largely consistent with 1-channel mechanisms having the largest relative decrease in integrated information due to anesthesia (Fig 4D). However, 2- and 3-channel mechanisms performed worse than 4-channel mechanisms ($\beta = -0.042$, $t(12) = -3.237$, $p = .007$, and $\beta = -0.017$, $t(12) = -3.156$, $p = .008$) despite having larger relative decreases in integrated information due to anesthesia, indicating that the reduction in 4-channel mechanisms, while smaller than that for 2- and 3-channel mechanisms, is more reliable. Meanwhile, the full multi-dimensional IIS outperformed integrated information of individual mechanisms and system-level integrated information ($\beta = 0.078$, $t(12) = 11.36$ compared to individual 1-channel mechanisms, and $\beta = 0.086$, $t(12) = 6.644$ compared to system-level integrated information, $p < .001$), implying that the structure of integrated information may reflect quantity of consciousness better than the simple summary provided by system-level integrated information. To rule out the possibility that the IIS performed better simply because it uses more coefficients to fit the data, we also performed model selection analyses by using logistic regression and comparing Akaike Information Criterion values. Even after penalizing complexity of the model, we still found the IIS to outperform system-level integrated information (S5 Text).

We also tested whether the reductions in integrated information were reliable across flies. This is important because in certain clinical contexts, such as traumatic brain injury, there may be no baseline measurements available, ruling out within-subject assessment. We conducted the decoding analysis, this time repeating leave-one-fly-out cross-validation at each of 8 wake-anesthesia epoch pairs (see Methods). We found that the trend of results for discriminating wakefulness from anesthesia, among mechanism sizes, was similar to within-fly classification, though accuracies and performance differences were overall reduced (Fig 5B). As before, classification accuracy varied depending on what measure was used ($\chi^2(5) = 4451$, $p < .001$). In contrast to the within-fly analysis, we found that (1) the IIS performed similarly to system-level integrated information ($\beta = 0.007$, $t(7) = 1.396$, $p = .206$), (2) 1-channel integrated

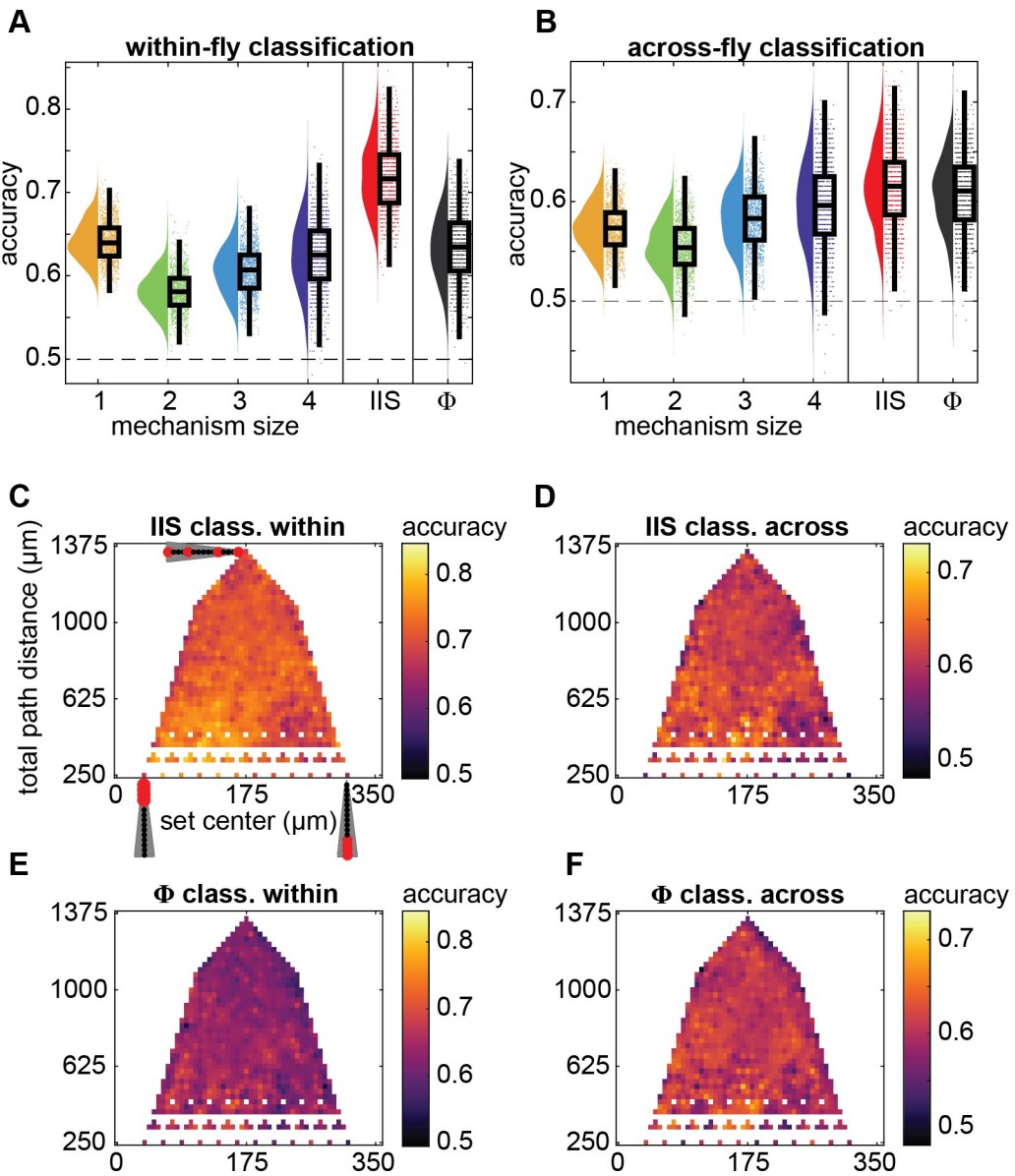

**Fig 5. Classification of wakeful vs. anesthetized conditions using mechanism-level φ, system-level Φ, or the integrated information structure (IIS: a set of φ's).** (**A**) Within-fly and (**B**) across-fly classification at each individual channel set using individual φ values for each mechanism size (orange, green, pale and dark blue are 1-, 2-, 3-, and 4-channel mechanisms, respectively; single-feature classification), when using the IIS (i.e. all mechanisms together, 15-feature classification, red), and when using Φ (single-feature classification, black). Individual points are classification accuracy of each channel set, after averaging accuracies across all mechanisms within the same mechanism size. Boxplots show median, 25th-75th percentiles, and whiskers are 1.5 interquartile below and above respectively. (**C-F**) Spatial map of classification accuracy (same format as in Fig 3). Within-fly (**C**) and across-fly (**D**) classification accuracies when using the IIS. Within-fly (**E**) and across-fly (**F**) classification accuracies when using Φ.

information only outperformed 2-channel integrated information (β = 0.018, $t(7)$ = 3.161, $p$ = .016), with (3) 4-channel integrated information achieving greater accuracies than 1-, 2-, and 3-channel integrated information (β = 0.023, $t(12)$ = 2.515, $p$ = 0.040, β = 0.042, $t(7)$ = 7.976, $p$ < .001, and β = 0.014, $t(7)$ = 6.250, $p$ < .001, respectively). As for within-fly classification, system-level integrated information attained similar performance to the highest performing

mechanism size, ($\beta = 0.011$, $t(7) = 1.711$, $p = .131$, compared to 4-channel integrated information). The pattern of reduced classification accuracy for smaller mechanisms suggests that the precise location of electrodes, or precise anatomical configuration, may not necessarily have been preserved across flies. Meanwhile, larger mechanisms may be less sensitive to the exact anatomical placement of channels.

## Integrated information structure reliably collapses globally across the brain

Finally, we tested if the reliability of using the IIS to distinguish wakefulness from anesthesia depended on spatial features (Fig 5C and 5D). Similar to the trends for the raw system-level integrated information values previously, we found significant trends between classification accuracy and channel set location. Classification accuracy increased as channel sets moved closer to the central brain (with channel 1 being the most central in the brain), for both within- and across-fly classification (Table 1), and slightly decreased as channels became more spaced out, also for both within- and across-fly classification. Thus, while the IIS collapsed throughout the brain, it was most reliable for central regions. These same trends were present for classification when using the system-level integrated information values (Fig 5E and 5F), though the trend of decreasing accuracy with more spaced out channels was not significant. Overall, using the IIS to discriminate level of consciousness in the fly brain yielded better classification accuracies, while maintaining the same spatial pattern of results as system-level integrated information. These results suggest that multidimensional measures may hold greater promise in distinguishing arousal states than more traditional single scalar value summaries of conscious level.

## Discussion

In this paper, we applied the measures derived from the Integrated Information Theory (IIT) of consciousness [30], one of the major quantitative theories of consciousness, to the neural recordings obtained from biological brains under two levels of arousal. We demonstrated the construction of integrated information structures (IIS), operationalised based on IIT 3.0, from real neural data to measure level of conscious arousal. We investigated how both system-level integrated information, the primary measure of conscious level put forward by IIT, and these information structures, consisting of a subset of the cause-effect structure (CES) proposed by

**Table 1. Dependence of regressands on channel set location and distance among channels.**

| Regressand | Location | | | Distance | | |
|---|---|---|---|---|---|---|
| | $\beta$[b] | $\chi^2(1)$[c] | $p$ | $\beta$[b] | $\chi^2(1)$[c] | $p$ |
| $\Phi$ Within | -9.16 | 44.30 | < .001 | -2.10 | 52.11 | < .001 |
| $\Phi$ Across | -5.76 | 16.77 | < .001 | -0.47 | 2.56 | .110 |
| IIS Within | -7.89 | 32.71 | < .001 | -2.09 | 51.05 | < .001 |
| IIS Across | -9.06 | 42.03 | < .001 | -0.90 | 9.31 | .002 |

Regressands were the classification accuracies reported in Fig 5C–5F. $\Phi$ Within: within-fly classification accuracy using $\Phi$, system-level integrated information. $\Phi$ Across: across-fly classification accuracy using $\Phi$. IIS Within: within-fly classification accuracy using IIS, integrated information structure. IIS Across: across-fly classification accuracy using IIS. Location: average location of channels in a channel channel set. Distance: sum of pairwise distances between channels within a channel set.

b $\beta$ from regressing $z$-scored classification accuracies; values are $\times 10^{-2}$

c The degree of freedom for all likelihood ratio tests was 1 (see Methods).

IIT as corresponding to the structure of consciousness, varied with change in level of arousal in the fly.

To distinguish conscious arousal states of human subjects, previous studies have employed other measures, inspired by IIT [41,8,42,43], on neural data. However, rigorous assessment of IIT ultimately requires assessing its proposed measures, not proxies thereof. So far, empirical testing of IIT has been lacking in this regard. Instead, research has focussed on comparing varying operationalizations of system-level integrated information with regards to theoretical requirements [44,33,37,45,46] or specific network architectures [40,47,48]. Meanwhile, there are relatively few papers on testing system-level integrated information as a measure of consciousness in neural data [49]. Further, to our knowledge only one paper has empirically investigated the notion of information structures, but in the context of the correspondence between the structures and conscious content rather than level of arousal [35].

Consistent with IIT's predictions, we found system-level integrated information to be reduced during anesthesia, and this was accompanied with the collapse of the information structures as reflected by loss of integrated information across all mechanism sizes. Further, we found that the collapse in the information structure during anesthesia was more reliable than the reduction in system-level integrated information, allowing us to classify wake from anesthesia with greater accuracy than using the scalar summary measure. Finally, we found that both the reduction in system-level integrated information and the collapse of the information structures were fairly uniform across all the channel sets which we considered, as was the reliabilities of their respective reduction and collapse. Overall, these results suggest significant recurrent interactions across the whole fly brain, contrary to a general view that the fly brain is largely feedforward, and demonstrate the utility of using information structures to assess level of arousal, over a scalar measure such as system-level integrated information.

## Global effect of anesthesia on system-level integrated information and the IIS

An influential view on the fly brain is that they are structured with largely feedforward and unidirectional synaptic connections, with possible exception in the central brain areas which have been identified as centers for integration [24,50,51,10]. From this view, we would expect minimal integrated information for peripheral regions and potentially greater integrated information for more centrally located channel sets, as system-level integrated information by design should be greater for those areas which have stronger recurrent connectivity as a whole (see S7 Text). There is however an emerging view that suggests that fly brains are densely connected in a hierarchical way much like mammalian brains [52]. This latter view implies that fly brains may be equipped with functionally recurrent and feedback computations like those of mammalian brains.

We found system-level integrated information to be slightly greater for more peripherally located channel sets (Fig 3A). Further, we found the decrease in system-level integrated information due to anesthesia, along with the collapse of the IIS to occur throughout the brain, regardless of location of the channel set or distance among channels in the set. Together, these results suggest that feedback interactions occur not only in the central areas of the fly brain, but also in more peripheral, sensory areas as well as across the whole brain. While this is in contrast to the general view of processing in the fly brain periphery being predominantly feedforward [53], feedback connections have indeed been reported in the fly brain (e.g.from the medulla to the lamina, and from the lamina to photoreceptors [54]). The finding of greater system-level integrated information in the peripheral, sensory processing, areas is also consistent with an indirect prediction of IIT that sensory areas are more important for consciousness [47], rather than higher order executive areas.

We acknowledge, however, potential limitations underlying our recordings and analyses. Firstly, it is conceivable that, due to the complexity of numerous brain structures in the centre the brain compared to the relative simplicity of fewer structures in the periphery [52], signals from a mix of many different structures may have cancelled each other out at the raw LFP level. Nonetheless, these central structures may have been more sensitive to the effects of anesthesia. Indeed, we found the effects of anesthesia on system-level integrated information and the IIS to be slightly more reliable for central channel sets (Fig 5C and 5E). Secondly, our method of discretising LFP voltages into binary states may not accurately represent the true space of real states of each of the channels, and also assumes equal probabilities of each state. Further, while IIT 3.0 focuses on moment-by-moment states, other methods, such as considering spectral power in time windows [55] may be more useful in describing the states of the channels, and so expanding IIT's framework to consider frequency domain data potentially is a promising avenue for future research [56]. Thirdly, we note that spurious high-order correlations can be found in partially observed multivariate systems and Markovian approximations of non-Markovian systems. These three limitations can be addressed through further investigation, especially with recordings at higher spatial resolutions than LFP, such as optical imaging or neuropixel probes [57], and expanding of IIT's theoretical framework.

## Why are 1-channel mechanisms more affected by anesthesia?

We encountered unexpected results with regards to 1-channel mechanisms. They had larger decreases in integrated information due to anesthesia (Fig 4C and 4D) and higher classification accuracy (Fig 5A) than larger mechanisms composed of 2, 3, or 4 channels. We had suspected that, as integrated information is supposed to measure information which is generated above and beyond separate parts, it would reflect the strength of long-range connectivity, which has been shown to be disrupted by isoflurane anesthesia, in humans, rats, and flies [25,26,36,58]. Given this background, we had expected that larger mechanisms, which are more likely to reflect long range connectivity, would be much more reduced and more reliable in classifying conscious states than 1-channel mechanisms.

We see two ways of interpreting this. First, if we consider 1-channel mechanisms as providing information to the rest of the channel set, then disrupting communication among individual channels inevitably leads to disruption of larger mechanisms. A second interpretation is that the large decrease in integrated information for 1-channel mechanisms may primarily reflect disruption of strong self-connections present during wakefulness, rather than communication with other channels. Having said that, we note that 1-channel integrated information is not a well-developed theoretical construct. In fact 1-channel integrated information isn't clearly defined for earlier versions of IIT and its approximations [28,33,44,45,59]. Specifically, integrated information for a mechanism is assessed by comparing the information it generates before and after imposing some disconnection among its parts. 1-channel mechanisms however cannot be split and compared in this manner. While IIT 3.0 specifically considers disconnections between a mechanism and its purview, and so some disconnection can always be imposed for any mechanism-purview combination, disconnections must still separate the mechanism into independent parts (each affecting their own independent purviews) [60], and thus the problem remains. In Fig 1G, we illustrate that the purviews of mechanisms A and B were simply themselves. In this example, imposing a disconnection on these self-connections seemed to result in a relatively large loss of information (compared to mechanism AB). While further investigation is necessary to understand our finding regarding 1-channel integrated information (e.g. such as 1-channel integrated information being potentially related to

autocorrelation; see S8 Text), our main results regarding the IIS are unaffected, as we verified that 1-channel mechanisms were not driving its classification performance (S6 Text).

### Role of system-level integrated information

In light of better classification accuracy of wake and anesthesia achieved by the computationally cheaper IIS, one might question the relevance of system-level integrated information in measuring conscious level. However, system-level integrated information plays key roles in IIT other than measuring level of consciousness.

Specifically, system-level integrated information is critical for two key roles. Firstly, it is used for identifying the "complex", the set of parts which maximise system-level integrated information [61–63]. Identification of the complex is critical for determining the boundaries of a system. Once identified, the CES generated by the complex is the "maximally irreducible conceptual structure", which is proposed to directly correspond to contents of consciousness.

Secondly, system-level integrated information is critical in identifying the ideal description of the system across spatial and temporal scales (e.g. individual neurons versus populations of neurons versus LFPs, or ideal sampling rate or time delay $\tau$), with the ideal description corresponding to the physical substrate of consciousness [48,64]. In the same vein, it can be used to identify e.g. the ideal function and/or threshold for binarizing states of the system (though we binarized voltages using the median to ensure equal entropy across channels and all epochs). The ideal description of the system is realised when system-level integrated information is maximal, and the IIS at that description is proposed to correspond to the experience of the system. These uses however require knowledge of all possible system elements, searching across many combinations of system elements, and searching across parameters for operationalising system states. Consequently, a proper, complete search remains infeasible for real neural data. We did however repeat our analyses at two different timescales, finding the same trend of results (see S2 Text).

### Differences between perturbation and observation in building the TPM

In order to compute the IIS, we built transition probability matrices (TPMs) and measured the information generated by the system when it is in a particular state. While ideally the TPM should be built by perturbing the system into all states and observing the immediate transition at the next timestep, this is something which is not currently achievable in intact brains. Thus, we built the TPMs by observing the natural, spontaneous evolution of time courses. Natural observation and perturbation can provide the same TPM if a few assumptions are met. First, the correct descriptions of the system must be identified (e.g. at the ideal spatiotemporal scale, and operationalization of states). Second, all states need to be reached during natural observations. When these assumptions are not met, perturbation becomes necessary to obtain a complete TPM.

Perturbation should also be used for setting "background conditions" [30], which is required to distinguish common inputs into system parts from truly integrated parts. Consider an example of two flies, where both flies are stimulated identically. Without taking into account the common stimulation, neural activity in one fly may correspond to and predict neural activity in the other fly, and so system-level integrated information computed from a TPM built from natural observation may not be able to indicate the presence of two separate systems. To avoid this, explicit perturbation (e.g. forcing stimulation to only one fly at a time) can be conducted to separate out the common stimulation.

## Applying IIT to loss of arousal in flies

We cannot be sure of the presence of consciousness in flies. Despite this uncertainty, we argue that the research program we are putting forward here is important and meaningful for several reasons. In particular, the fundamental approach of inferring consciousness in animals is to search for behavioural and physiological similarities between ourselves and the animal in question. With sufficiently strong behavioural and physiological similarities, we may at some point consider that the weight of evidence favours attributing consciousness to that particular animal [65,66]. In this context, there is accumulating evidence to suggest that flies indeed have varying levels of consciousness [20,21,10]. There is even some evidence to suggest similar psychological processes in flies as in humans, such as attention [16,21,67], memory [13–15] and feature binding [19]. Further similarities have been found for other insects, such as perception of illusory contours, metacognition, false memory, and long-term planning in bees [68–71].

Assessing the validity of IIT's constructs using recordings from the fly brain provides key advantages, compared to testing in humans. Firstly, using multi-electrode methods provides high quality population neural signals in both time and space, unaccessible with any non-invasive measures available in humans. Further, the small brain of the fly allows us to obtain recordings covering successive layers of visual processing simultaneously, from the retina to the central brain. Secondly, given how the computational cost of computing system-level integrated information and the associated information structures grows exponentially with the number of channels being considered simultaneously [39], the smaller number of neurons in the fly brain, compared to mammalian brains ($10^5$ compared to $10^8$ for mice and $10^{11}$ for humans [72–74]), provides a system where computing these measures across a large majority of neurons is more feasible. The smaller brain size of the fly has already allowed for detailed imaging of neural circuits across large portions of the fly brain [75,76]. Detailed knowledge of connections among neurons can in the future help inform computation of IIT constructs, e.g. in reducing the set of disconnections to search through when computing integrated information of mechanisms or system-level integrated information. Thirdly, the fly brain is already extensively used as a model of anesthetic loss of consciousness, and various observed molecular mechanisms of anesthesia, such as decreased action potential amplitudes [77,78], and effects on network dynamics such as reduced feedback connectivity [12,26,79], seem to be conserved across species. Further, fly brains appear to share graph-theoretical characteristics with mammalian brains [52] as well as cellular mechanisms [80], and fly LFPs share similarities with human electroencephalographic recordings [81,82]. Taken together, the fly serves as a useful model for investigating the constructs of IIT.

## Conclusion and future outlook

Our work opens up several future directions for empirically assessing mathematical approaches to consciousness, especially for IIT. It will be also important to test the generality of our finding across different modulations of consciousness, such as considering graded levels of anesthesia or sleep, as well as across datasets from different systems, such as in the more complex mammalian brain. Even without presuming consciousness for a given system, applying such approaches can inform biology, such as bringing focus to feedback interactions in a system which is largely considered feedforward. While we investigated the use of information structures in determining level of arousal, IIT links these structures more directly to contents of consciousness. As flies can demonstrate complex behaviors such as attentional selection [83] it would be interesting future research to see if the structures of consciousness in flies that can be reasonably inferred from behaviors would correlate with the structures of integrated information as in humans [35].

## Methods

### Experimental procedure

As the data have been published in [36], here we detail methods directly relevant to the current manuscript. Thirteen female laboratory-reared *Drosophila melanogaster* flies (Canton S wild type, 3–7 days post eclosion) were collected under cold anaesthesia and glued dorsally to a tungsten rod.

Linear silicon probes with 16 electrodes (Neuronexus Technologies) were inserted laterally into the fly's eye. Probes had an electrode site separation of 25 μm. Recordings were made using a Tucker-Davis Technologies multichannel data acquisition system with a 25 kHz sampling rate. Isoflurane was delivered from an evaporator onto the fly through a connected rubber hose. Actual concentration near the fly body was either 0 vol% (awake condition) or 0.6 vol% (isoflurane condition). Flies in the awake condition responded to air puffs by moving their legs and abdomen, but were rendered inert under the isoflurane condition. Importantly, they regained responsiveness when isoflurane was subsequently removed, ensuring that flies were alive during the anesthesia recording.

The experiment consisted of two blocks: one for the 0% isoflurane (air condition, followed by one for the isoflurane condition. Each block started with a series of air puffs, followed by 18 s of rest, 248 s of visual stimuli, another 18 s of rest, and finally a second series of air puffs. Isoflurane was administered immediately after completion of the first block (i.e. after the last air puff), and flies were left for 180 s to adjust to the new concentration before beginning the second block. We used data obtained in the 18 s period between the end of the first series of air puffs and the beginning of the visual stimuli.

### Local field potential preprocessing

LFPs were downsampled to 1000 Hz from their original sampling rate of 25 kHz. Downsampled LFPs were bipolar re-referenced by subtracting neighbouring electrodes, resulting in 15 signals which we refer to as "channels". The 18 s of data for each condition was split into 2.25 s segments, giving 8 epochs of 2250 time-samples each. We removed line noise at 50 Hz using the function rmlinesmovingwinc.m function of the Chronux toolbox (http://chronux.org/; [84]) with three tapers, a window size of 0.75 s, and a step size of 0.375 s. Finally, we binarized voltages by taking the median voltage for each channel across all time-samples within a 2.25 s epoch, and then converting each time-sample in the epoch to 'on' if the voltage for that time-sample was greater than the median, and 'off' otherwise (for the effect of binarization threshold, see S1 Text).

### IIS computation

Data processing for computing the IIS and system-level integrated information was conducted using Python 3.6.0 in MASSIVE (Multi-modal Australian ScienceS Imaging and Visualisation Environment), a high-performance computing facility. We calculated the measures using PyPhi (version 0.8.1; [39]), publicly available from https://github.com/wmayner/pyphi. Complete details of all the calculations can be found in [30,39].

To compute the IIS, transition probability matrices (TPMs) describing how the set of channels transition from one state to another across time are required. To estimate these, we first select a set of $n$ channels of interest, for which there are $2^n$ possible states. For each channel in the set, we computed the empirical probability of being "on" at time $t+\tau$ given the state of the system at time $t$. This gives a $2^n \times n$ matrix (i.e. a "state-by-channel" matrix), which can then be directly fed to the PyPhi toolbox [39]. We use $\tau = 4$ ms as $\tau$ which is too small will not

capture causal interactions which maximise integrated information, based on known physiology of synaptic interactions [85]. A comprehensive search across $\tau$ is infeasible due to computational cost (but see S2 Text for repeated analyses also at $\tau = 2$ ms and 6 ms).

The state-by-channel TPM is used in IIT 3.0, which assumes that there are no instantaneous interactions among the channels (i.e. the "conditional independence" assumption). In other words, the state of some channel being '1' or '0' at some time point is not affected by the state of other channels at the same time point. This assumption is reasonable for classical physical systems, but may not hold when not all units' interactions are considered (e.g. when there is common input to the system). As it is infeasible to obtain a full description of all parts and interactions of intact brains, this is a limitation of the current IIT 3.0 operationalisation of integrated information (note however that the issue is dealt with and resolved for a previous version of IIT by explicitly incorporating conditional dependence among system parts [33,37]).

We computed the state-by-channel TPMs for every possible, 4-channel subset out of the 15 channels (15choose4 = 1365 channel sets), repeating this procedure for each fly and epoch (obtaining one TPM per fly and 2.25s epoch). We selected 4 channels as this gave a reasonable balance between system-level integrated information and the IIS's strength of being a multivariate measure and their weakness of exponentially growing computation cost with system size [39].

To compute the IIS and system-level integrated information for a given set of 4 channels at a given epoch, we submitted its associated transition probabilities to PyPhi. Conceptually, PyPhi finds distances between the probability distribution of transitions specified by the full system with that of the disconnected system (Fig 1G and 1H). As there are $2^n$ possible states for a set of $n$-channels (16 states for 4-channels), we computed a set of 15 integrated information values (the IIS) and one system-level integrated information value for every state. Within each epoch, we first computed the within-epoch state-weighted average [40]. For the comparison of integrated information values between wakefulness and anesthesia, we further averaged these values across the 8 epochs.

In Fig 1G and 1H, we explained system-level integrated information ($\Phi$) as the sum of distances between cause and effect probability distributions specified by the full and (minimally) disconnected system (i.e. the full CES and the disconnected CES). Distances for each mechanism are weighted by the mechanism's $\varphi$ value, as $\varphi$ is the "earth" which is being moved from the full to the disconnected system (consequently, the distance is weighted by the smaller $\varphi$ out of the full CES and disconnected CES). Any differences in $\varphi$ values between the full and disconnected system, such as for mechanism AB in Fig 1G and 1H, are "moved" to the maximally uninformative distributions (EMD$^c$ and EMD$^e$; red asterisks between Fig 1G and 1H). This weighted summation is depicted in between Fig 1G and 1H, in the solid box.

## Classification analysis

To assess the reliability of the effects of anesthesia on the IIS, we conducted classification analysis, which allows us to compare the multivariate IIS (15-features) with single mechanism integrated information (1-feature) and system-level integrated information (1-feature) values. We trained and tested SVMs for each channel set using LIBLINEAR (using default options, i.e. L2-regularized L2-loss support vector classification (dual) [86]) at two levels: a) classifying epochs within each fly (within-fly classification, repeated for each fly), and b) classifying flies at each trial (across-fly classification, repeated at each epoch).

For each measure (integrated information of individual mechanisms, IIS, or system-level integrated information), we conducted nested leave-one-pair-out cross-validation [87,88]. At each outer validation, we conducted an inner-cross-validation procedure on 7 epoch-pairs

(within-fly classification; an epoch-pair consists of one wakeful and one anesthetized epoch) / 12 fly-pairs (across-fly classification; a fly-pair consists of one epoch of each wakefulness and anesthesia from the same fly), where we trained SVMs on 6 epoch-pairs / 11 fly-pairs at a time, and validated performance on the remaining epoch-pair / fly-pair. Training features (integrated information values or system-level integrated information values) were each $z$-scored before training, and testing features were $z$-scored using the mean and standard deviation of the training set. This was repeated at different cost hyperparameters ($2^{-50}$ to $2^{50}$, in steps of powers of 10).

We then trained a SVM on all 7 epoch-pairs / 12 fly-pairs used in the inner-cross-validation, repeating the $z$-scoring procedure, at the cost hyperparameter value which gave the greatest validation performance (in cases of tie conditions, we took the lower cost value), and tested the overall classifier on the remaining epoch-pair / fly-pair. For the majority of validations (~74% for within-fly classification), the lowest cost of $2^{-50}$ was selected as the cost value. This process was repeated for each fly / epoch (within-fly classification / across-fly classification), and we averaged across repeats to obtain a final classification accuracy for the channel set and measure. For accuracy of mechanisms with a given size, we report averaged accuracies across all mechanisms with the given size (e.g., we report 1-channel mechanism accuracy as the average accuracy across all 1-channel mechanisms).

## Statistical analyses

We used linear mixed effects analysis (LME; [89,90]) to test for significant differences. LME allows us to account for within-fly correlations among channel sets and avoid averaging across either channel sets or flies. Thus we always included random intercepts for fly and the interaction between fly and channel set as random effects, unless otherwise specified. To test for statistical significance of an effect, we employed likelihood ratio tests, where we compared the log-likelihood of the full model with a model with the effect of interest removed. As the likelihood ratio statistic is $\chi^2$ distributed when one model is nested in another, we report the likelihood ratio statistic with the associated degrees of freedom ($\chi^2(d.o.f.)$) corresponding to the difference in number of coefficients between the full model with the model with the effect of interest removed, as well as the corresponding $p$-value. To conduct pairwise comparisons (e.g. to compare 1-channel to 2-channel integrated information), we limited the effect of interest to two levels at a time and report the associated regression coefficient. As $p$-values associated with these regression coefficients were very small and potentially do not reflect the true degrees of freedom, we report the coefficients along with "classical" group-level $t$-tests (conducted after averaging across channel sets to obtain a single value per fly or, for across-fly classification, per epoch).

We first employed LME to compare system-level integrated information, $\Phi$, between wakefulness and anesthesia, using the following model (in Wilkinson notation [90]):

$$\Phi \sim condition + (1|fly) + (1|fly : set) \tag{1}$$

Where condition is level of conscious arousal (wake or anesthesia; dummy coded to be treated as a categorical variable), fly is individual flies (treated as a nominal variable), and set is channel set (treated as a nominal variable). In Table 2, we summarize the amount of variance explained in each model as well as the intercepts for random effect.

To test for a relationship between system-level integrated information ($\Phi$) values and channel set location or total path distance among channels, we regressed system-level integrated information values onto channel set location and distance among channels:

$$\Phi \sim condition + location + distance + location : condition + distance : condition + (1|fly) + (1|fly : set) \tag{2}$$

**Table 2. Linear mixed effects model fit (adjusted $R^2$) and standard deviation ($SD$) of random effects.**

| | $R^2$ | SD | | |
|---|---|---|---|---|
| **Random effect** | | $+ (1|f)^{\#}$ | $+ (1|f{:}n)^{\wedge}$ | $+ (1|n)^{\&}$ |
| $\Phi \sim c$ | .489 | 0.011 | $3.185 \times 10^{-11}$ | |
| $\Phi \sim c + l + d + l{:}c + d{:}c$ | .493 | 0.011 | $1.524 \times 10^{-11}$ | |
| $\varphi \sim c + s + c{:}s$ | .412 | $5.95 \times 10^{-3}$ | $4.07 \times 10^{-3}$ | |
| $\Delta\varphi \sim s$ | .372 | 0.235 | 0.121 | |
| $a_W \sim F$ | .476 | 0.019 | | |
| $a_A \sim F$ | .309 | 0.020 | | |
| $a_{\Phi W} \sim l + d$ | .562 | | | 0.683 |
| $a_{\Phi A} \sim l + d$ | .513 | | | 0.702 |
| $a_{\varphi W} \sim l + d$ | .555 | | | 0.686 |
| $a_{\varphi A} \sim l + d$ | .535 | | | 0.694 |

Model specifications are described in detail in Methods. $\Phi$: system-level integrated information. $c$: level of arousal (wake/anesthesia). $l$: channel set location. $d$: sum of pairwise distances between channels within a channel set. $\varphi$: (mechanism-level) integrated information. $s$: mechanism size. $\Delta\varphi$: ratio of wakeful to anesthetized integrated information for mechanism-level integrated information. $a_W$: within-fly classification accuracy. $F$: feature used for classification (categorical variable; individual 1-, 2-, 3-, 4-channel mechanisms, 1 feature; IIS, 15 features; or system-level integrated information, 1 feature). $a_A$: across-fly classification accuracy. $a_{\Phi W}$: within-fly classification accuracy using system-level integrated information. $a_{\Phi A}$: across-fly classification using system-level integrated information. $a_{\varphi W}$: within-fly classification accuracy using the IIS. $a_{\varphi A}$: across-fly classification accuracy using the IIS.

\# Random intercept for effect of fly.

^ Random intercept for interaction between fly and channel set.

& Random intercept for channel set.

Where "location:condition" and "distance:condition" denote interaction terms between channel set location and condition, and distance among channels and condition, respectively. We describe the relationship between $\Phi$ and channel set location or distance among channels by reporting regression coefficients from $z$-scored $\Phi$ values in addition to the significance of the effect of location from the likelihood ratio test.

To compare integrated information ($\varphi$) values of the IIS between wakefulness and anesthesia and among mechanism orders, we used the model:

$$\varphi \sim condition + size + condition : size + (1|fly) + (1|fly, set) \tag{3}$$

Where size is mechanism size (1, 2, 3, or 4, dummy coded to be treated as a categorical variable). The number of observations among mechanism sizes differed due to each order having a different number of possible mechanisms (4, 6, 4, and 1, respectively for 1-, 2-, 3-, and 4-channel mechanisms). The term "condition:size" denotes an interaction between level of conscious arousal and mechanism size.

To compare the differential effects of anesthesia among mechanism sizes (breaking down the significant interaction between condition and size in the previous LME), we used the model:

$$\Delta\varphi \sim size + (1|fly) + (1|fly : set) \tag{4}$$

Where $\Delta\varphi$ is the ratio of wakeful to anesthetized integrated information.

When comparing classification accuracy across flies across the different feature types (i.e. 1-, 2-, 3-, and 4-channel $\varphi$, the IIS, and $\Phi$), classification accuracy was not nested within fly, thus

we only included random intercepts for each channel set:

$$accuracy \sim feature + (1|set) \tag{5}$$

Where feature was dummy coded to be one of 1-, 2-, 3-, or 4-channel φ, the full IIS, or Φ.

To test for a relationship between classification performance and channel set location or distance among channels, we regressed accuracies onto the two spatial features:

$$accuracy \sim location + distance + (1|set) \tag{6}$$

Where accuracy is classification accuracies, averaged across flies or epochs (for within-fly and across-fly classification, respectively). As for the relationship between Φ and the spatial features, we describe the relationship between accuracies and the spatial features by reporting regression coefficients on $z$-scored accuracies in addition to the significance of the effect of location from the likelihood ratio test.

## Supporting information

**S1 Text. Effect of anesthesia is consistent at different binarization thresholds.**
(PDF)

**S2 Text. Effect of anesthesia is consistent using different timesteps.**
(PDF)

**S3 Text. "Disconnection" through statistical noising.**
(PDF)

**S4 Text. Effect of anesthesia on system-level integrated information for each fly.**
(PDF)

**S5 Text. IIS best predicts wakeful vs. anesthesia states.**
(PDF)

**S6 Text. 1-channel mechanisms do not drive classification performance of the IIS.**
(PDF)

**S7 Text. Recurrent activity is required for greater system-level integrated information.**
(PDF)

**S8 Text. Relation between 1-channel mechanisms and autocorrelation.**
(PDF)

## Acknowledgments

We would like to thank Andrew Haun, Leonardo Barbosa, and William Mayner for their comments on our manuscript. This work was supported by computational resources provided by the Australian Government through MASSIVE under the National Computational Merit Allocation Scheme.

## Author Contributions

**Conceptualization:** Angus Leung, Naotsugu Tsuchiya.

**Data curation:** Dror Cohen.

**Formal analysis:** Angus Leung.

**Funding acquisition:** Bruno van Swinderen, Naotsugu Tsuchiya.

**Investigation:** Angus Leung, Dror Cohen.

**Methodology:** Dror Cohen, Bruno van Swinderen, Naotsugu Tsuchiya.

**Project administration:** Naotsugu Tsuchiya.

**Resources:** Dror Cohen, Bruno van Swinderen.

**Software:** Angus Leung.

**Supervision:** Naotsugu Tsuchiya.

**Validation:** Angus Leung.

**Visualization:** Angus Leung.

**Writing – original draft:** Angus Leung.

**Writing – review & editing:** Angus Leung, Dror Cohen, Bruno van Swinderen, Naotsugu Tsuchiya.

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
