## [Decision Letter · Decision Letter 0]

30 Jul 2020

Dear Mr. Leung,

Thank you very much for submitting your manuscript "Integrated information structure collapses with anesthetic loss of conscious arousal in Drosophila melanogaster" for consideration at PLOS Computational Biology.

As with all papers reviewed by the journal, your manuscript was reviewed by members of the editorial board and by several independent reviewers. In light of the reviews (below this email), we would like to invite the resubmission of a significantly-revised version that takes into account the reviewers' comments.

These issues are quite major, and concern both the validation of the results, as well as their biological relevance. At this stage I cannot exclude that a revised version in which the technical and statistical aspects have been addressed, but with an unclear evidence of the novel insight into the biological system could be still judged not suitable for publication in PLOS Computational Biology, and would be offered publication in PLOS One. We want to offer you the possibility of working on both aspects, but still being transparent on this uncertainty, and aim to have the least of your time lost. 

We cannot make any decision about publication until we have seen the revised manuscript and your response to the reviewers' comments. Your revised manuscript is also likely to be sent to reviewers for further evaluation.

Sincerely,

Daniele Marinazzo

Deputy Editor

PLOS Computational Biology

Daniele Marinazzo

Deputy Editor

PLOS Computational Biology

Reviewer's Responses to Questions

**Comments to the Authors: **

Reviewer #1: Review: Integrated information structure collapses with anesthetic loss of conscious arousal in Drosophila melanogaster

Summary: 

The authors apply Integrated Information Theory (IIT) to local field potentials (LFPs) recorded from fruit flies during wakefulness and general anesthesia. The primary result is a decrease in integrated information and the collapse of information structures during anesthesia compared to wakefulness. 

Applications of IIT to neuroimaging data are rare, and those that exist are mostly based on earlier versions of the theory (e.g., “IIT2.0”). The current work is novel in that it applies the newest version of the theory (“IIT3.0”) to neuroimaging data, and it represents an important contribution to the field. 

One reason that there have been few previous applications of IIT3.0 is because it is not possible to perform the computations in large networks. For this work, the authors propose a heuristic analysis that balances theoretical motivations with practical considerations. Moreover, the authors explicitly acknowledge the limitations and caveats of their analysis. I find the work to be technically sound. 

For these reasons, I support its publication in PLoS Computational Biology. Below I list a few issues that could benefit from some clarification in a minor revision. 

Abstract:

Line 29: as you point out in the discussion, your TPM is calculated by observation and not perturbation. You should not describe such interactions as causal (also other places in the Results/Methods).

Results:

Your integrated information structure (IIS) seems very similar to the cause-effect structure (CES) of IIT3.0. When you introduce the idea, can you elaborate on how the IIS is different from the CES (is it because of the various practical approximation required?). 

Lines 168-171: In IIT, this should just be the cause-effect structure. For the MICS (the M stands for maximally, not minimally), you first need to search for the set of elements that maximizes Phi. 

Methods:

I only found the value of tau in the figure caption. It would be good to include in the methods text. Also, how was the value of tau = 4ms selected? It would be good to know that similar values of tau (e.g, 2ms or 6ms) give similar results. 

Were separate TPMs for each 2.25s epoch? Or were the epochs somehow combined to create a single TPM? This was not clear for me. 

Discussion:

Regarding the claim that the IIS is more reliable than Phi for assessing level of consciousness. A potential pitfall here is that there is nothing to stop non-integrated systems from having many large small phi values. For example, if you took 4 channels from one fly, and four channels from another fly, then you could have lots of mechanisms with non-zero small phi, but the whole system would not be integrated (and the two flies presumably not jointly conscious). That there were in fact two systems is something you would only see with Phi. In your situation, it works out okay because all the systems you analyze are integrated to some degree, but it would be good to address this.

Reviewer #2: This paper claims to show that system-level integrated information is reduced in the fly brain during anesthesia, and that structures of integrated information collapse in the anesthetized state. These results are interpreted as confirming the integrated information theory of consciousness.

As it stands, I am not convinced by this analysis, for several reasons. 

The first reason is conceptual, and centers on the difficulty of inferring the presence, absence, or character of consciousness in the fly brain. To begin, it is far from clear whether flies are conscious at all during waking states. Moreover, if flies are conscious during waking states, then it still is impossible to infer what the phenomenology of that consciousness is like. This is a major issue for the reported analyses, because the integrated information theory of consciousness - on which the reported analyses are based - begins with the phenomenology of human consciousness. In particular, the theory begins with the observation that human consciousness forms an integrated whole, i.e. that percepts across all of our sensory modalities are simultaneously available to us. This phenomenological observation serves as the basis for the mathematical structure of the theory, and those are the mathematical structures used here to analyze the fly brain. Problematically, we cannot know if this integrated phenomenology is present in flies, even if we assume that flies have some form of subjective experience. And there is reason to doubt that flies experience an integrated perceptual whole, because it is not even clear that all vertebrates experience integrated perception. For example, extensive experiments on the visual systems in frogs suggest that frogs’ visual perception is not integrated as ours is (see Chapter 1 of Milner and Goodale’s The Visual Brain in Action). And, while there is some evidence of multi-sensory integration in the mushroom bodies of insect brains (see, e.g., Farris 2002, “Evolution of insect mushroom bodies: old clues, new insights”) and in the ventrolateral lobes of a few insects (see, e.g., Anton et al 2011, “Brief predator sound exposure elicits behavioral and neuronal long-term sensitization in the olfactory system of an insect”), in general it’s thought that sensory processing in insect brains is not as integrated as it is in vertebrate brains (see Chittka and Niven 2009, “Are Bigger Brains Better?”). In summary, even if the integrated information theory of consciousness is true, it is still unclear whether we should expect a priori that flies are conscious during waking states, that their experience is integrated during waking states, that they are unconscious under anesthesia, and that their unconsciousness under anesthesia is due to a collapse of information integration. 

Aside from these conceptual issues, there are also a number of major analysis issues in this paper. The first problem, which the authors note in the Discussion, is that the recorded local field potentials are discretized through a simple binarization at the median. The transition probability matrices which are used to calculate integrated information are then estimated based on this discretization. While such binarization has been employed elsewhere, for e.g. in computing the Lempel-Ziv complexity of brain signals, it is unclear whether such a binarization would yield accurate results in the estimation of integrated information, particularly for a continuous, nonlinear signal like a local field potential. Presumably, the structure of state transitions in the fly brain is far more complex than can be captured through a simple binarization. Moreover, it is unclear how the authors estimate the transition probability matrix of the “statistically disconnected” fly brain. As the authors note, the calculation of integrated information requires comparing the transition probability matrix of a full system to the transition probability matrix of that same system, once that system has been severed at its weakest informational link. A “statistical” estimate of the behavior of a system following a network cut is possible only for linear, Gaussian variables. But, because the dynamics of the fly brain are presumably nonlinear, it is, by definition, impossible to estimate the transition probability matrix of the disconnected fly brain without actually physically severing neural connections in the fly brain - and doing so for all possible partitions or bipartitions of the brain. Moreover, even if the authors’ method for statistically estimating the transition probability matrix of the disconnected fly brain were valid, they do not perform this for all possible cuts of the system, as they note on lines 183-188. What guarantee do we have then that their estimate is close to the ground-truth? Considering the number of heuristics used in this paper’s methods - namely, binarization, statistical estimation of the disconnected transition probability matrix, and searching through only a sub-sample of partitions - I would need to see much more evidence that these heuristics approximate the ground-truth in at least a simulation of a complex, nonlinear system before I am convinced that these heuristics approximate the ground-truth in empirical recordings from a complex, nonlinear system.

Finally, I would need to see more evidence that the effects reported here are not simply driven by changes to the entropy of the flies’ local field potentials under anesthesia. There is evidence that reports of changes to neural information transfer under anesthesia may in fact be driven by drops in local entropy (see Wollstadt et al 2017, “Breakdown of local information processing may underlie isoflurane anesthesia effects.”). In other words, it may be that the mechanisms of integration/communication are unaffected by anesthesia, and that anesthesia instead disrupts how much information is available to be integrated/communicated. Reports of a breakdown of information transfer or communication under anesthesia may therefore be erroneous. In light of Wollstadt et al’s findings, I would like to see that the results reported in this paper are not driven by changes to entropy. This may be done by, for e.g., normalizing estimates of integrated information by estimates of entropy, and showing that these normalized estimates are likewise reduced under anesthesia, or by statistically showing that variance in integrated information cannot be explained by variance in entropy. On a related note, I wonder if a disruption to local entropy may be driving the authors’ finding that 1-channel mechanisms are the most strongly affected by anesthesia. 

Based on these issues, I recommend rejection or otherwise major revision of this paper.

Reviewer #3: ## Summary

In this paper, Leung et al apply measures inspired by a version of integrated

information theory (IIT 3.0) to LFP data of awake and anesthetised flies. The

main result is that system-level integrated information \\Phi and

mechanism-level integrated information \\phi decrease under general anesthesia.

Furthermore, the authors show that classifiers built on the whole set of \\phi

(an integrated information structure, IIS) is better at predicting whether

flies were anesthetised than classifiers built on \\Phi or \\phi alone.

The main contribution of this paper is the first application of a measure

inspired by IIT 3.0 to neural data. I would not say that this is IIT 3.0,

because the theory relies on many assumptions (Markovianity, full

observability, causal perturbations) that do not hold in this analysis, and

involves calculation steps that have not been taken here (the authors mention

this in the Discussion). The paper is well written, and data has been made

publicly available. Although the results seem interesting, I have a few

methodological concerns that must be addressed before the quality and

significance of the paper can be fully judged.

## Major comments (most important first)

1) Statistical reporting

My main concern with the paper is the inappropriateness of the statistical

reporting. Most prominently, the paper reports some incredibly small p-values

(e.g. t-values of -85), which I believe are strongly misleading. When used in

this kind of studies, p-values typically refer to group-level comparisons, and

not to the p-value of coefficients of the model (with repeated measures, etc).

The method used by the authors leads to excess degrees of freedom that are

radically underestimating the p-values of the results. As an example, in the

first reported LME model in L.207 the actual number of degrees of freedom is

much closer to 13 (due to the correlation between \\Phi values in the same fly)

than to 35488.

In general, the authors should report more comprehensive statistics, and avoid

misleading p-values. This includes, but is not limited to:

- Performing t-tests with 13 degrees of freedom of quantities of interest (e.g.

 \\Phi, within-fly class. accuracy) averaged across epochs and channel sets.

- Showing more descriptive statistics of \\Phi, like mean \\Phi and effect size

 of anesthesia for each fly.

- Reporting standard deviation of random effects in the LMEs.

- Reporting R^2 for the LMEs.

2) Control for model size in IIS classification

One of the central results of the paper is that classifiers using IIS as

features are better than classifiers that use \\Phi or \\phi. However, while from

Fig 5 it visually seems to be the case (at least for within-fly

classification), the paper does not provide conclusive evidence that IIS are

significantly better, since it is to be expected that classifiers built on IIS

may perform better just by virtue of having more trainable parameters.

While the paper does report some p-values related to the classifiers (L.312 and

below), I suspect it may be another instance of the issue with the p-values I

pointed out above. To have more convincing evidence that the IIS is indeed

statistically preferred as a predictor, the authors should use standard model

selection techniques for each fly/channel set, confirm that the IIS model is

always preferred (using standard criteria like AIC or BIC), and report the

results explicitly.

3) Single-mechanism \\phi

The high values and strong effects obtained for single-mechanism \\phi are

indeed, as the authors point out, very unexpected. While these do not

contribute to system-level \\Phi directly, they do form an important part of the

IIS and therefore play a role in the paper's main results. Since they are

certainly "not a well-developed theoretical construct", the authors should

elaborate more on what they are, what they mean, and how they contribute to the

results.

For example, the authors could make sure that this single-mechanism \\phi does

not contribute to the IIS classification accuracy by repeating the analyses

with IIS made of only sets of two or more mechanisms.

4) Nature of \\Phi measures computed

Throughout the paper, the authors refer to the interactions being quantified by

their analysis as "causal." This is not true, and it gives a false impression

of similarity with the theory in Oizumi 2014. The authors should draw a clear

boundary between the theory as presented in Oizumi 2014 (which does very

explicitly deal with causal interventions) and the present analysis, which does

not. The authors should also clarify (or at least mention) the differences

between the measures applied here and IIT 3.0 in the Introduction.

On a related topic, despite the clear and informative explanation of the

computation of \\phi, the computation of \\Phi is not at all clear in either the

text, the figure, or the caption. What does it mean to "best approximate the

MICS"? Is this the exact formula given in Oizumi 2014, or some ad-hoc

approximation? Please clarify.

## Minor comments (most important first)

- In several figures (Fig 4A,C, colour bars in Fig 3) the y-axis has a

 non-uniform spacing, which creates a potentially misleading perception of the

 data. Please change.

- There were several analysis details that were in figure captions or in the

 Methods section, and should be more explicitly stated upfront -- for example,

 the value of \\tau and the fact that all measures are averaged across system

 states.

- What is the justification for \\tau = 4ms?

- In Fig 1C, why isn't the TPM a 4x4 matrix, with 4 future states?

- The description of the classifiers could be more clear, for example, by

 including the number of features and data points used in each classifier. In

 particular: are all the classifiers except the one based on IIS using just a

 single predictor?

- I find it unusual that, among mechanisms of size 2, 3 or 4, the wake/anest

 \\phi ratio is reduced with larger mechanism size, but the classification

 accuracy is increased (while I would intuitively expect these to be

 positively correlated). Do the authors have any explanation for this?

- In general, the overall style of figures could be improved (i.e. adjust

 number of significant figures, uppercase letters in labels, font size, etc).

- The format of the references (i.e. title caps vs sentence caps) should be

 consistent.

- There is a '??' symbol in the caption of Fig. 1 (L.796).

**Have all data underlying the figures and results presented in the manuscript been provided?**

Reviewer #1: Yes

Reviewer #2: None

Reviewer #3: Yes

PLOS authors have the option to publish the peer review history of their article (what does this mean?). If published, this will include your full peer review and any attached files.

Reviewer #1: No

Reviewer #2: No

Reviewer #3: No
---

## [Decision Letter · Decision Letter 1]

21 Dec 2020

Dear Mr. Leung,

Thank you very much for submitting your manuscript "Integrated information structure collapses with anesthetic loss of conscious arousal in Drosophila melanogaster" for consideration at PLOS Computational Biology. The reviewers and the editors appreciated the effort you made towards the revised version. There is still an outstanding issue regarding the binarization of the time series that would need more re-examination. 

Sincerely,

Daniele Marinazzo

Deputy Editor

PLOS Computational Biology

Daniele Marinazzo

Deputy Editor

PLOS Computational Biology

[LINK]

Reviewer's Responses to Questions

**Comments to the Authors:**

Reviewer #1: the authors have satisfactorily addressed all of my points.

Reviewer #2: This draft of this paper is much improved over the previous draft. In particular, I appreciate the authors’ far more nuanced discussion of what their results imply for our understanding of the fly brain and for consciousness more broadly. I also appreciate the authors’ clearer discussion of how they computed transition probability matrices, and how those were statistically disconnected for all possible system cuts. My concerns about these points have been satisfactorily addressed.

I am still, however, concerned with the validity of computing these matrices from binarized time-series, given that all of the results reported in this paper rest on the validity of this approach. Given that this binarization is used to compute transition probability matrices, the authors’ approach essentially assumes that a local field potential can only enter two “relevant” states, with an equal probability of being in one state or the other (though, on this point, I appreciate their demonstration that their results are consistent across different binarization thresholds). As I said in my previous review, this is a problematic assumption for a continuous, non-spiking process like a local field potential. A much more rigorous approach to the same problem was taken by Hudson et al, “Recovery of consciousness is mediated by a network of discrete metastable activity states,” PNAS (2014), where discrete transition probability matrices were estimated from local field potential recordings using k-means clustering on principal components estimated from the data. Moreover, given the results reported therein (i.e., that a local field potential can spend more time in some states than in others), a simple binarization at the median for both awake and anesthetized signals (which assumes equal time spent in each state) cannot capture the actual state transitions of the system. The authors’ simulation results using a nonlinear autoregressive process do alleviate my concerns along these lines somewhat, and as such I strongly recommend including that analysis in the supplement. But, I would still like to see either a more rigorous discretization approach (for e.g. the one taken by Hudson et al, using k-means clustering of principle components estimated from local field potential data), or at least a more detailed discussion of the limitations of the simple binarization used here.

Reviewer #3: I would like to congratulate the authors on a clear improvement of the manuscript. Related to my previous review, statistical reporting has substantially improved (Tables 1 and 2 are especially valuable), and the extra tests and model metrics provide much more information for readers. I am satisfied with the updated discussion of statistical causality, the role of the TPM and its relation with Oizumi2014.

I have recommended the paper for acceptance, although I very strongly suggest the authors consider two further points which have not been addressed so far:

- Most importantly, on the topic of 1-channel mechanisms: the authors should mention (and possibly explore further in future work) the relation between 1-channel \\phi and single-channel auto-correlation. Could a change in auto-correlation between conditions explain (some of) the observed results?

- I find the claims about feedforward systems rather overstated: it is possible to have spurious high-order correlations in multivariate systems when they are partially observed, or when a non-Markovian system is approximated through a Markovian assumption (as is the case here). In this sense, the simulation the authors provided as a reply to Reviewer #2 does not really address the reviewer's concerns, since it is not a non-linear, non-Markovian, or partially observed system.

As a minor comment, the argument that the authors didn't consider \\tau > 6ms because of limited data is rather poor -- increasing \\tau by 1ms only reduces the amount of data by 1 sample (judging by the 1kHz sampling frequency), which is not a huge reduction. The authors are free to keep this argument if they like, but I suspect informed readers will find it unpersuasive.

**Have all data underlying the figures and results presented in the manuscript been provided?**

Reviewer #1: None

Reviewer #2: Yes

Reviewer #3: Yes

PLOS authors have the option to publish the peer review history of their article (what does this mean?). If published, this will include your full peer review and any attached files.

Reviewer #1: No

Reviewer #2: No

Reviewer #3: No
---

## [Editor Report · Decision Letter 2]

18 Jan 2021

Dear Mr. Leung,

We are pleased to inform you that your manuscript 'Integrated information structure collapses with anesthetic loss of conscious arousal in Drosophila melanogaster' has been provisionally accepted for publication in PLOS Computational Biology.

Best regards,

Daniele Marinazzo

Deputy Editor

PLOS Computational Biology

Daniele Marinazzo

Deputy Editor

PLOS Computational Biology

---

## [Editor Report · Acceptance letter]

19 Feb 2021

PCOMPBIOL-D-20-00825R2 

Integrated information structure collapses with anesthetic loss of conscious arousal in * Drosophila melanogaster*

Dear Dr Leung,

I am pleased to inform you that your manuscript has been formally accepted for publication in PLOS Computational Biology. Your manuscript is now with our production department and you will be notified of the publication date in due course.

With kind regards,

Alice Ellingham
